# Exploring the Well-Being of Slovenian and Serbian Sport Science Students during the COVID-19 Pandemic of Summer 2022

**DOI:** 10.3390/sports11020040

**Published:** 2023-02-06

**Authors:** Brigita Banjac, Ivana M. Milovanović, Saša Pišot, Radenko M. Matić, Stevo Popović, Sandra S. Radenović, Patrik Drid

**Affiliations:** 1Faculty of Sport and Physical Education, University of Novi Sad, 21000 Novi Sad, Serbia; 2Western Balkan Sport Innovation Lab, 81000 Podgorica, Montenegro; 3Institute for Kinesiology Research, Science and Research Centre Koper, 6000 Koper, Slovenia; 4Faculty for Sport and Physical Education, University of Montenegro, 81400 Nikšić, Montenegro; 5Faculty of Sport and Physical Education, University of Belgrade, 11000 Belgrade, Serbia

**Keywords:** sport, health science, physical education, university students, COVID-19 pandemic, mental health, well-being, PERMA

## Abstract

The COVID-19 pandemic profoundly changed everyday life of social actors, which inferred mental health and well-being concerns. As students of health-related studies tend to adapt better to difficult circumstances, in this study, we explored the effect of the pandemic on sports science students’ well-being during the summer of 2022. The research was conducted in Slovenia and Serbia. The sample comprised *n* = 350 students. The PERMA-Profiler, a 15-item self-reported questionnaire, was adapted to assess well-being across five elements: positive emotions, engagement, relationships, meaning, and accomplishment. Data were collected with a questionnaire through the summer of 2022 (from May to July) and analyzed in SPSS, AMOS 26.0. The results revealed normal functioning (M = 7.72, SD ± 1.38) for the overall well-being of the students. Although all dimensions indicated high scores, relationships (M = 7.95, SD ± 1.63), meaning (M = 7.76, SD ± 1.69), and engagement (M = 7.73, SD ± 1.36) rated the highest. Furthermore, the instrument was acceptable, as the confirmatory factor analysis showed adequate reliability based on Cronbach’s alpha (15 items, α = 0.94) and strong internal correlations between the PERMA dimensions. This study contributes to the previously published research, emphasizing the positive responses and successful coping of sports science students in times of complex situations, such as the COVID-19 pandemic.

## 1. Introduction

The reality of what people knew until December 2019 has profoundly changed from then until now. The emergence of the COVID-19 pandemic [1] caused various restrictive measures for reducing the spread of the virus, among which the most obvious were movement control, physical distance among individuals, and curfew. Although they had a preventive function to preserve public health, they changed the behavior and life of individuals [2,3]. In this way, these measures necessitated people’s adaptation to new public health and social circumstances. From the perspective of the university student population, these serious measures (such as closed universities, online learning, and isolation physical distance among individuals) resulted in lifestyle changes [4] and mental health and well-being difficulties [5,6,7,8,9].

Well-being and mental health are important factors for individuals’ normal functioning. Although there are many different tools for measuring well-being, there is no ideal profile: “Different profiles may be more or less adaptive for different people at different times, depending on their personality, history, and social context” [10]. However, we applied positive psychology, which attempts to measure well-being across five domains from a positive-based standpoint. It can be traced back to Martin E. P. Seligman, who wrote that “Psychology is not just the study of disease, weakness, and damage; it also is the study of strength and virtue. Treatment is not just fixing what is wrong; it also is building what is right” [11]. Thus, based on the complexity of the well-being construct, one promising approach is the multidimensional model from positive psychology [10].

Physical activity brings many positive benefits to the individual [12] and has an important role in protecting mental health [13,14]. During the pandemic, physical activity can alleviate adverse mental health effects [15] and improves mood [16] and resilience among students [17]. Namely, it helps in developing coping abilities within these challenging conditions. In favor, one of the students’ most commonly used strategies was physical activity [18]. On the other hand, students who had a decrease in their physical activity level reported lower well-being [19]. Therefore, it plays an important role in the fight against the epidemics’ consequences. Furthermore, besides many positive health gains, it helps to build certain habits and absorb socially acceptable norms and values. For instance, students from the sports faculties reported that they exercised regularly through the emergency measures, demonstrating that they have defined health-related routines in their daily practices [4].

Apart from the mental health concerns, the students had to face an unknown length of time of higher uncertainty and anxiety regarding their studies and career [20,21,22]. Although the pandemic led to uncertainty about the future, the sport science students’ intolerance level of uncertainty was moderate [23], which indicates that those types of students tend to have higher levels of adaptation. Moreover, sport science students compared to musicians tend to have more skills to face the challenges and obstacles that appear along their path [24]. Indeed, sport science students compared to students in other study programs showed more positive and better coping with the difficult circumstances caused by the pandemic [4]. Therefore, we assume that sport science students adapted well and have normal functioning in their everyday lives during the first waves of the COVID-19 pandemic.

In brief, the COVID-19 pandemic has already been present for more than one and a half years in everyday life. Therefore, university students are exposed to those extraordinary social circumstances for extended periods of time, which, based on the literature, in the beginning, resulted in mental health and well-being challenges. Even though there are few studies exploring sport science students, we aimed to explore how the students adapted to these circumstances and, more precisely, to assess the well-being of Slovenian and Serbian sports science students during the summer of 2022.

## 2. Materials and Methods

### 2.1. Design

This study was part of a more comprehensive research project called “Everyday life of students in the extraordinary social circumstances of the COVID-19 pandemic: a comparative study”, conducted during 2022 in two countries: Slovenia at the University of Primorska and Serbia at the University of Novi Sad, including altogether ten faculties. Its goal was to compare the impact of the pandemic on university students’ everyday life and their way of adapting to these extreme social circumstances (the COVID-19 pandemic) among the student population of sports and other (social, natural, and applied) sciences.

### 2.2. Participants

The study included a sample of *n* = 350 (male 52%, female 48%) sports science students from Slovenia and Serbia. They studied at the University of Primorska, Faculty of Health Sciences, and at the University of Novi Sad, Faculty of Sport and Physical Education. Their ages ranged between 19 and 30 years (M = 22.98, SD ± 2.19). Detailed descriptions of the participants are listed in Table 1.

The sampling processes for reducing the number of participants from the initial total sample size of the above-mentioned research project for the present study are shown in Figure 1.

Even though the above-mentioned research project included a larger sample size (*n* = 1060), for the purpose of this study, we applied a few relevant exclusion criteria. At the beginning, we excluded *n* = 52 due to disagreement to participate in this study. Although there were 1008 questionnaires eligible for analysis, looking at the obtained data, we noticed a certain homogeneity in the answers of sports science students. Hence, we decided to present sports science students as an autonomous group within the general student population included in this field research. Additionally, given that the “Sports” journal is by definition more focused on sports activities, we decided to single out students of sports sciences for the needs of the study. Therefore, the sample size is limited to *n* = 354 and includes only sports science students from the Faculty of Health Sciences (University of Primorska) and the Faculty of Sport and Physical Education (University of Novi Sad). Lastly, we considered *n* = 350 for the analyses, as this number of participants completed all the mandatory questions (without missing data and dropping out) of the PERMA-Profiler questionnaire.

### 2.3. Instruments

This study was carried out using a questionnaire designed for the above-mentioned research project’s data collecting on the topic of the impact of the pandemic on students’ everyday life and their way of adaptation to these extreme social circumstances (the COVID-19 pandemic). In line with the research problem and the project objectives, we applied both quantitative and qualitative methodologies in the research project. However, for the purpose of this study, we analyzed only part of the questionnaire. The benefit of the survey lies in the ability to measure subject behavior, attitudes, opinions, emotions, and intentions based on the respondents’ answers.

The questionnaire was divided into the following sections:The first included socio-demographic questions about students, namely gender, age, country and university, graduation course, sports experience (at least three consecutive years with competitions), and students’ view of the pandemic;The second part followed, which contained the PERMA-Profiler questionnaire for measuring well-being.

Martin Seligman’s well-being theory is based on positive psychology’s goal to increase people’s flourishing. To achieve that, he separated five dimensions of the well-being construct: positive emotion, engagement, relationships, meaning, and accomplishment. None defines well-being by itself, but each contributes to the meaning of the latent concept [25]. Butler and Kern made these above-mentioned dimensions measurable with the PERMA-Profiler survey. The English version was evaluated for its psychometric measures across a large international sample (*n* = 31.966), resulting in a 15-item questionnaire. In addition, eight items for assessing negative emotions were added. The validation study of this measurement tool indicated adequate reliability and validity of the scale. [10]. It has become one of the most used tools for evaluating well-being [26] and has been translated into more than 20 languages. In addition, the questionnaire is being used across diverse cultural contexts [27,28,29,30,31,32,33]. The reliability of subscales of the PERMA-Profiler was assessed with an internal consistency reliability coefficient (Cronbach’s alpha) using the data from this study, representing an estimate of 0.94.

To assess multi-dimensional well-being, we employed the PERMA-Profiler, a 15-item scale questionnaire developed for adults. Its measures Seligman’s five pillars of well-being: (P) positive emotions, (E) engagement, (R) relationships, (M) meaning, and (A) accomplishment. Each of these dimensions contains three items. In addition, an 11-point Likert scale ranging from 0–10 was adapted for the response rate, where higher scores indicated better well-being and vice versa [10]. The composite scores from each of the three items per construct were averaged. The overall well-being score is the mean of the 15 items [34]. As a scoring system, we classified the results as the following: very high functioning (9 and above), high functioning (8–8.9), normal functioning (6.5–7.9), sub-optimal functioning (5–6.4), and languishing (below 5) [35]. The questions were translated from English to Slovenian and Serbian languages for better understanding. That is, all the questions were translated from English into Slovenian and Serbian languages. Then, they were independently back translated into English by second and third translators. At this stage of the study, the authors included four members in the expert committee (one expert from the field of Psychology of Sports, two from the Sociology of Sports, and one from English in Sports Sciences). In the next step, the text was translated back to the original language (English), and the expert committee reviewed to finalize the translation. The back-translated versions were then sent back to original authors, who confirmed its accuracy. Further, pre-testing of the translated version of the scale was conducted, and the expert committee agreed on the final version of the scales. Lastly, validation process of the scale in terms of its reliability and validity is explained in detail in the method and results section.

### 2.4. Procedures

This cross-sectional study had a descriptive design. Data were collected during the summer of 2022 (from May to July) across students from the University of Primorska and from the University of Novi Sad. Even though the instrument was a self-administered questionnaire, the participants received a description of the study’s aim, an explanation of the used terminology, and instructions for the filling process. The printed version of the questionnaire was disseminated to students after their classes, and the online version was distributed, during or after the students’ classes at the faculties, across online platforms through the help of the authors’ social network, which was followed by a snowballing approach where we asked participants to forward the link to their student peers. The link for the survey was created across the 1KA platform (https://www.1ka.si/d/en/about/general-description (accessed on 13 May 2022)), namely an open-source application that enables services for online surveys developed by the Centre for Social Informatics, Faculty of Sciences, University of Ljubljana. In addition, data were processed and managed by the General Data Protection Regulation (GDPR). This research was conducted following the ethical standards of the Helsinki Declaration. Accordingly, before completing the survey, the students were asked to indicate that they understood the aim, method, and purpose of the study and that they gave consent for agreeing to participate voluntarily in this research without providing any personal information (name, birth date, and contact information). They could withdraw consent to participate at any time. Furthermore, their data are anonymous and used only for scientific research purposes. Ethical approval prior to data collection was obtained at the Faculty of Sport and Physical Education, University of Novi Sad (No. 47-12-12/2021-1).

### 2.5. Statistics

Data were analyzed using the IBM SPSS (version 26.0, IBM, Armonk, NY, USA) and the AMOS 26.0 program. First, we calculated descriptive statistics such as mean, standard deviation, and frequency. Furthermore, following the PERMA scoring system, scores for each factor (positive emotion, engagement, relationships, meaning, accomplishment) were calculated as the average scores of the three items from the survey that made up the one factor. Furthermore, the overall well-being was calculated as the average score of all factors (fifteen items).

In addition, structural equation modeling was used to test the proposed model. Testing the model and paths among PERMA variables used two fit indices—root mean square error of approximation (RMSEA) and comparative fit index (CFI). In addition, based on recommendations by Hu and Bentler [36], Cronbach’s alpha was used to measure internal consistency for each sub-scale and overall well-being, composite reliability (CR), and average variance extracted (AVE). In this process, we applied the cut-off criteria by Hu and Bentler [36]: CMIN/DF—terrible > 5, acceptable > 3, excellent > 1; CFI—terrible < 0.90, acceptable < 0.95, excellent > 0.95; RMSEA—terrible > 0.08, acceptable > 0.06, excellent < 0.06.

## 3. Results

### 3.1. Description of The Participants

The questionnaire of PERMA-Profiler was answered by 350 participants; of those, 182 (52.0%) were male, and 168 (48.0%) were female students. Their age ranged from 19–30 years (M = 22.98, SD = 2.19). The largest percentage of the students were from Serbia: *n* = 245 (70%). Information about their graduation course, sports experience, and their views of the pandemic is presented in Table 1.

### 3.2. PERMA-Profiler Results

As presented in Figure 2, the results revealed that the general score indicates normal functioning among students, accordingly, within all sub-domains of well-being. The highest means were obtained for the factors of relationships, meaning, and engagement.

Specifically, overall well-being had a mean score of 7.72 with SD ± 1.38; the positive emotion factor had M = 7.70 with SD ± 1.73; the engagement factor had M = 7.73 with SD ± 1.36; the relationships factor had M = 7.95 with SD ± 1.63; the meaning factor had M = 7.76 with SD ± 1.69; and the accomplishment factor had a mean score of 7.43 with SD ± 1.56. It is important to note that the overall sub-domains (P, E, R, M, and A) scores of the individual items were averaged. Therefore, the score for overall well-being is the mean of all (fifteen) items calculated.

### 3.3. Confirmatory Factor Analysis (CFA)

Factorial validity of scales, which considered reliability and validity of the measurement model, is presented in Figure 3.

### 3.4. Internal Consistency

Reliability based on Cronbach’s alphas in Table 2 showed a strong internal consistency for overall well-being (15 items, α = 0.94) and positive emotions (3 items, α = 0.93), and a moderate internal consistency for relationships (3 items, α = 0.80), meaning (3 items, α = 0.87), and accomplishment (3 items, α = 0.80). On the other hand, it was relatively low for the engagement factor (3 items, α = 0.54). Therefore, the reliability analysis showed acceptable values of Cronbach’s alpha for four constructs (>0.80), as recommended by Nunnally and Bernstein [37], while only the dimension of engagement was below this criterion, which in line with the scales’ validation samples findings [10]. Moreover, Table 2 provides information that the correlations between the five PERMA dimensions were positively significant and mainly strong. If the participant reported higher positive emotion, they also tended to have a higher level of engagement (r = 0.87, *p* < 0.01), relationships (r = 0.89, *p* < 0.01), meaning (r = 0.85, *p* < 0.01), or accomplishment (r = 0.76, *p* < 0.01).

Further, all indicators of CR were greater than or equal to 0.60 (0.60–0.92), which satisfied the criteria of Bagozzi and Yi [38]. Four constructs of AVE fulfilled the criteria of being greater than 0.50, as suggested by Fornell and Larcker [39], but one construct (engagement) showed a lesser value (0.36). As seen in Table 3, the measurement model showed an acceptable fit (x^2^ = 395.81, df =160, x^2^/df = 2.47, CFI = 0.94, and RMSEA = 0.06). The CFI values revealed excellent fitting of the model, and the value of RMSEA is 0.06, which was proposed as acceptable by Hu and Bentler [36].

The criterion values of factor loading satisfied the suggestion of Kaiser [40], where 14 factor loading values were greater that 0.4 (>0.56), while one item in the scale of engagement (e3) was below this criteria (0.26). In general, it revealed that the measurement model fit well with the empirical research data (Table 4).

## 4. Discussion

Based on a positive psychology framework [11,25] and few previously published articles on compatible topics [4,24], our main interest was in university students’ well-being after two years of the COVID-19 pandemic outbreak. Therefore, our aim was to investigate and describe how the students adapted to these circumstances and, more precisely, to assess the well-being of Slovenian and Serbian sports science students during the summer of 2022.

The results revealed normal functioning (M = 7.72, SD ± 1.38) for the overall well-being of the students, which aligns with our assumption that Slovenian and Serbian sport science students’ snapshot of well-being in summer 2022 indicates normal functioning. However, based on the literature the accompanying extraordinary social circumstances, the pandemic directly impacted daily lives [3,4,21] and brought many mental and well-being challenges, especially in the early phases of the pandemic. There is also evidence that people showed better resilience through time compared to the early stages of the pandemic [41]. As well as some key aspects of this study, students had higher positivity about the pandemic as time went on [9]. Thus, their psychological well-being during the second wave of the pandemic was better than during the first one [5]. This can be related to the fact that they had a prolonged period in a stressful environment to learn how to adapt as well as gain more knowledge about the COVID-19 pandemic.

Furthermore, it should not be ignored that several studies suggest that physical activity, one of the main topics of the sport science students’ curriculum, positively correlates with well-being [42]. If the students decreased their physical activity level, they received lower scores on the well-being scale [19]. Thus, it is a great tool for enhancing or maintaining mental health [43] and mood improvement [16], especially in uncertain circumstances, such as the COVID-19 pandemic.

During the first wave of the pandemic, it was shown by comparing students from “health-related“ study programs to others that the former are less sensitive to changes in habits [4]. Furthermore, sport science students during the pandemic had a medium intolerance level of uncertainty, which also indicates that their behaviors and perceptions are moderate regarding the unknown factors about the future [23]. Moreover, they were physically active and maintained average scores in healthy lifestyle behaviors [44].

Moreover, results showed that all dimensions of the overall well-being indicated high scores, and relationships (M = 7.95, SD ± 1.63), meaning (M = 7.76, SD ± 1.69), and engagement (M = 7.73, SD ± 1.36) rated the highest. As we can see, the most important factor was relationships. Seligman also indicated that being social is one of the most successful forms of adaptation [25]. In addition, the most identified supporters during the pandemic were parents and friends [6]. Additionally, the level of well-being was positively correlated with students’ relationships [45] and emotional support [46]. Further, cooperating and helping others were positive experiences during the pandemic [9]. On the other hand, students during the lockdown period expressed loneliness and indicated a need for support in the form of meetings and better communication [47]. In addition, meaning can support individuals through the pandemic [48], and engagement has a positive association with students’ career and mental well-being and career competencies, while the opposite is true with burnout [21]. Furthermore, the instrument was acceptable, as the confirmatory factor analysis showed adequate reliability based on Cronbach’s alpha (15 items, α = 0.94) and strong internal correlations between the PERMA dimensions. Based on the literature [10,29,49], these measures can be interpreted as a relatively stable, acceptable instrument among the university student population in the public health context.

We can mention several practical implications of this research. Although the PERMA scale is considered to be still in its developmental stages, this research shows that it is relatively stable, making it an acceptable instrument among the university students population in the public health context of COVID-19 during the summer of 2022. Although for the purposes of this manuscript, we only indicated the results obtained among sports sciences students, we believe that this research is an extension of previously published, compatible empirical research. The importance of the results was obtained from comparing students of different educational profiles [4]. Like some already published studies [50], in our research, it was noticeable that the adjusted PERMA scale demonstrated its ability to cut across demographic factors with a certain degree of stability.

This study, as an extension of the previously published research, emphasizes the positive responses and successful coping of sport science students in periods of extraordinary public health circumstances, such as the COVID-19 pandemic in both countries during the third wave of the pandemic. However, our exploration did not result in conclusions-based causality. Follow-up studies are needed for measuring the further waves of the pandemic, including students from different study courses and with a bigger sample size. Additionally, instrument-measuring factors related to students’ everyday life and coping skills should be included. In summary, we attempted to explore how the students adapted to the difficult circumstances due to the COVID-19 pandemic at the third wave, and more precisely, we investigated the well-being of Slovenian and Serbian sports science students during the summer of 2022. The results revealed normal functioning for their overall well-being. Although all dimensions indicated high scores, the relationships, meaning, and engagement factors were rated the highest. This research contributes to the previously published research, emphasizing the positive responses and successful coping of sports science students in times of difficult and challenging situations, such as the COVID-19 pandemic.

## Figures and Tables

**Figure 1 sports-11-00040-f001:**
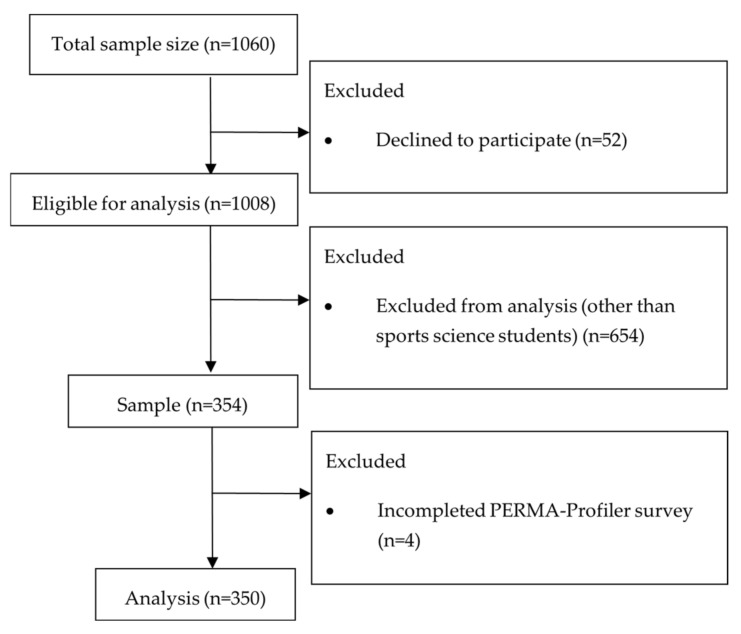
The sampling process.

**Figure 2 sports-11-00040-f002:**
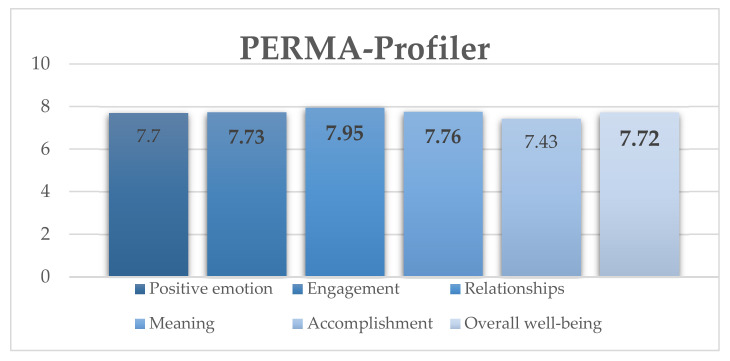
Mean scores on the PERMA-Profiler survey.

**Figure 3 sports-11-00040-f003:**
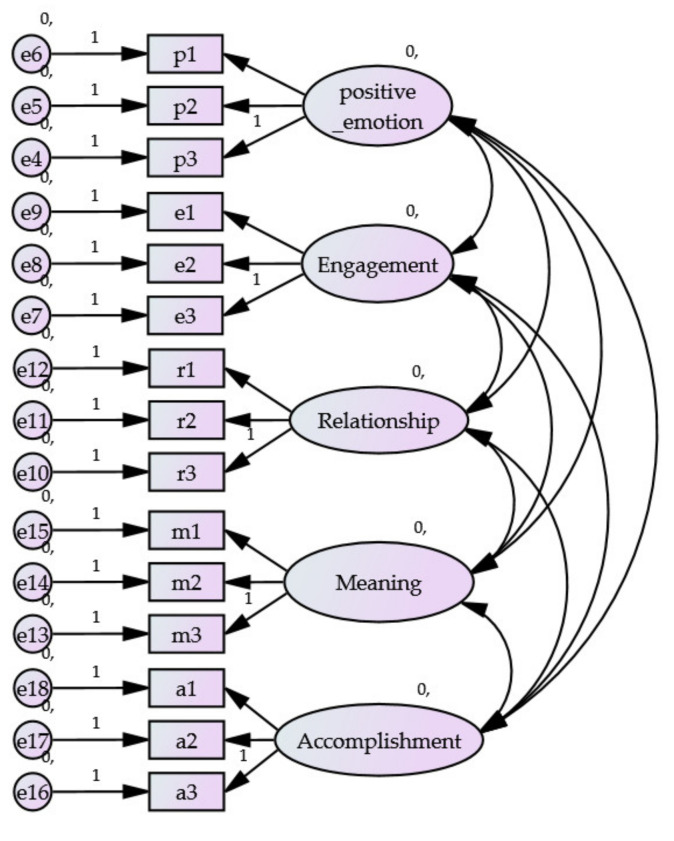
PERMA Model.

**Table 1 sports-11-00040-t001:** Demographic characteristics of the participants.

Variables	Demographic Characteristics	*n*	%
Gender	Male	182	52.0
	Female	168	48.0
Age (years)	19	6	1.7
	20	19	5.4
	21	64	18.3
	22	83	23.7
	23	69	19.7
	24	36	10.3
	25	30	8.6
	26	14	4.0
	27	11	3.1
	28	9	2.6
	29	4	1.1
	30	5	1.4
Country	Slovenia	105	30.0
	Serbia	245	70.0
Course	BSc	263	75.1
	MSc	75	21.4
	PhD	12	3.4
Sports experience	I am currently an athlete	98	28.0
	I used to be an athlete	192	54.9
	I was not an athlete	60	17.1
Students’ view of the pandemic	Crisis	164	46.9
	Opportunity	171	48.9
	Other	15	4.3

**Table 2 sports-11-00040-t002:** Means (M), Standard Deviations (SD), Cronbach’s Alpha, Composite Reliabilities (CR), Average Variance Extracted (AVE), and Correlations of PERMA dimensions.

Dimensions	M	SD	α	CR	AVE	r				
						P	E	R	M	A
Positive emotion (P)	7.70	1.73	0.93	0.92	0.80	**0.90**				
Engagement (E)	7.72	1.36	0.54	0.60	0.36	0.87 **	**0.60**			
Relationship (R)	7.95	1.63	0.80	0.81	0.59	0.89 **	0.74 **	**0.77**		
Meaning (M)	7.76	1.69	0.87	0.87	0.70	0.85 **	0.99 **	0.83 **	**0.84**	
Accomplishment (A)	7.43	1.56	0.80	0.80	0.58	0.76 **	1.01 **	0.71 **	0.92 **	**0.76**
Overall well-being	7.72	1.38								

Note: ** *p* < 0.01. Bold values represent a coefficient of multiple correlation.

**Table 3 sports-11-00040-t003:** Fit indices.

Measure	Est.	Threshold	Interpretation
CMIN	395.81	-	-
DF	160.00	-	-
CMIN/DF	2.47	Between 1 and 3	Excellent
CFI	0.94	>0.95	Acceptable
RMSEA	0.06	<0.06	Acceptable

Note: Applied cut-off criteria by Hu and Bentler (1999) [36].

**Table 4 sports-11-00040-t004:** Descriptive Statistics and CFA Item Statistics.

Variables	M	SD	Skewness		Kurtosis	Factor Loading	Error Term	SMCs
Positive emotion								
p1	7.54	1.82	−0.10		1.15	0.87	0.01	0.76
p2	7.82	1.89	−1.10		1.25	0.91	0.10	0.83
p3	7.75	1.84	−0.97		0.68	0.90	0.10	0.82
Engagement								
e1	7.41	1.89	−0.73		0.70	0.70	0.10	0.49
e2	7.83	1.78	−0.83		0.38	0.73	0.09	0.54
e3	7.94	1.99	−1.36		2.60	0.26	0.11	0.07
Relationship								
r1	7.75	2.02	−0.88		0.38	0.59	0.11	0.35
r2	8.14	2.01	−1.29		1.55	0.81	0.11	0.66
r3	7.96	1.77	−1.04		1.33	0.88	0.09	0.77
Meaning								
m1	7.58	1.88	−0.86		0.45	0.89	0.10	0.79
m2	8.02	1.87	−1.15		1.58	0.81	0.10	0.66
m3	7.70	1.94	−0.98		1.06	0.80	0.10	0.64
Accomplishment								
a1	6.97	2.00	−0.61		0.63	0.77	0.11	0.60
a2	7.35	1.80	−0.75		0.36	0.78	0.10	0.61
a3	7.98	1.72	−0.94		0.83	0.72	0.09	0.52

Note: CFA, confirmatory factor analysis; SD, standard deviation; SMC, squared multiple correlation.

## Data Availability

Not applicable.

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
