# Peer review of "Exploring the Well-Being of Slovenian and Serbian Sport Science Students during the COVID-19 Pandemic of Summer 2022"

_sports, 2023, doi:10.3390/sports11020040_

Round 1

Reviewer 1 Report

1) Why only sport students are selected as participants? Please clarify in the Participants.

2) What is sample size calculation? Also, what were inclusion and exclusion criteria?

3) What about the reliability and validity of the instrument?  

4) What are the practical implications for this study?

Good luck

Author Response

Reviewer 1

Comments and Suggestions for Authors

  1. Why only sport students are selected as participants? Please clarify in the Participants.
  • Dear reviewer,

      Thank you for the remark. This manuscript is part of a more comprehensive research, which included 1060 students of different educational profiles. Given that the "Sports" journal is by definition more focused on sports activities, we decided to single out students of sports sciences for the needs of the manuscript. At the same time, we explained the research project in the 2.1. “Design” section, and the data sampling process in the 2.2. "Participants" section, as you suggested through a time line as Figure 1.

  1. What is sample size calculation? Also, what were inclusion and exclusion criteria?
  • The sampling processes including the size calculation and the inclusion and exclusion criteria are explained in more detail in 2. “Materials and Methods” section of the paper.

  1. What about the reliability and validity of the instrument?
  • Thank you for your comment. The instrument validation process can be found in the following literature: Butler and Kern (2016). In brief, the English version was tested for its psychometric measures across a large international sample (N=31.966) and resulted in a 15 items questionnaire. In addition, 8 items for assessing negative emotions were added. The validation study of this measurement tool indicated adequate reliability and validity of the scale. (Butler and Kern, 2016). Therefore, based on the previously published research across many countries and sample using both the 23 item instrument and the shortened version of 15 items, we decided to implicate the sorter version of the questionnaire to our study (Avsec et al, 2022; Žibrat et al. 2021, Avsec et al., 2022; Umucu et al., 2020; Wammerl et al., 2019; Ryan et al., 2019; Varga et al., 2022). Furthermore,
  • Reliability of subscales of the PERMA-Profiler was tested using an internal consistency reliability coefficient. Thus, reliability based on Cronbach’s alpha is described in the 3. Results section, and especialy in 3.4. “Internal consistency” section. The reliability analysis showed acceptable values for four constructs (>0.80), as recommended by Nunally and Bernstein (1994), while the only dimension of engagement was below this criterion, which is in line with the scales’ validation samples’ findings (Butler and Kern, 2016). Moreover, Table 2 provides information that the correlations between the five PERMA dimensions were positively significant and mainly strong.

Literature mentioned in the answer:

  • Butler, J.; Kern, M.L. The PERMA-Profiler: A brief multidimensional measure of flourishing. International Journal of Wellbeing 2016, 6, 1-48, doi:10.5502/ijw.v6i3.526
  • Avsec, A.; Kavčič, T.; Kocjan, G.Z. Psychology of Subjective Well-Being; Ljubljana University Press, Faculty of Arts: Slovenia, 2022.
  • Žibrat, K. Validation of the PERMA questionnaire. In Proceedings of the 22nd Psychology Days in Zadar, Zadar, 2021.
  • Avsec, A.; Eisenbeck, N.; Carreno, D.F.; Kocjan, G.Z.; Kavčič, T. Coping styles mediate the association between psychological inflexibility and psychological functioning during the COVID-19 pandemic: A crucial role of meaning-centered coping. Journal of contextual behavioral science 2022, 26, 201-209
  • Umucu, E.; Wu, J.-R.; Sanchez, J.; Brooks, J.M.; Chiu, C.-Y.; Tu, W.-M.; Chan, F. Psychometric validation of the PERMA-profiler as a well-being measure for student veterans. Journal of American College Health 2020, 68, 271-277,doi:10.1080/07448481.2018.1546182.
  • Wammerl, M.; Jaunig, J.; Mairunteregger, T.; Streit, P. The German Version of the PERMA-Profiler: Evidence for Construct and Convergent Validity of the PERMA Theory of Well-Being in German Speaking Countries. Journal of Well-Being Assessment 2019, 3, 75-96, doi:10.1007/s41543-019-00021-0
  • Ryan, J.; Curtis, R.; Olds, T.; Edney, S.; Vandelanotte, C.; Plotnikoff, R.; Maher, C. Psychometric properties of the PERMA Profiler for measuring wellbeing in Australian adults. PLOS ONE 2019, 14, e0225932, doi:10.1371/journal.pone.0225932.
  • Varga, B.A.; Oláh, A.; Vargha, A. A magyar nyelvű PERMA Jóllét Profil kérdőív megbízhatóságának és érvényességének vizsgálata. Mentálhigiéné és Pszichoszomatika 2022, 23, 33-64, doi:10.1556/0406.23.2022.001
  • Nunnally, J.C.; Bernstein, I.H. Psychometric theory 3rd. ed.; McGraw-Hill: New York, 1994

  1. What are the practical implications for this study?
  • Thank you for your constructive comments. We can mention several practical implications of this research.Although the PERMA scale is considered to be still in its developmental stages, this research shows that it is relatively stable, making it an acceptable instrument among the university students’ population in the public health context of COVID-19 during the summer of 2022.Although for the purposes of this manuscript, we only indicated the results obtained among students of sports sciences, we believe that this research is an extension of previously published empirical research. The importance of the results obtained from comparing students of different educational profiles (Pišot et al., 2022).Like some already published studies (Magare et al., 2022), in our research, it was noticeable that the adjusted PERMA scale has demonstrated its ability to cut across demographic factors with a certain degree of stability. This study, as an extension of the previously published research, emphasizes the positive responses and successful coping of sport science students in periods of extreme circumstances, such as the COVID-19 pandemic in both countries during the third wave of the pandemic.

Literature mentioned in the answer:

  • Pišot, S.; Milovanović, I.M.; Katović, D.; Bartoluci, S.; Radenović, S.S. Benefits of active life in student experiences during COVID-19 pandemic time. Frontiers in Public Health 2022, 10, doi:10.3389/fpubh.2022.971268)
  • Magare, I.; Graham, M.A.; Eloff, I. An Assessment of the Reliability and Validity of the PERMA Well-Being Scale for Adult Undergraduate Students in an Open and Distance Learning Context. International Journal of Environmental Research and Public Health 2022, 19, 16886, doi:10.3390/ijerph192416886

Reviewer 2 Report

The research addresses a relevant question and the authors have analysed the data and produced a report on these data. The sample size is sufficiently large to warrant interest. However, the paper could be developed more.

a) the literature on psychological states and traits experienced during COVID-19 is comprehensive. The authors need to do a review of this literature, and inform the readers on what they did to see what research had been conducted. This is a comprehensive revision.

b) The authors need to develop a data analysis strategy for what they did. They investigate the validity of measures; which is a good, but do not make a case as to why this should be done and what the impact of this means on the scales themselves and their validity in this context. This could be an important part of the work. 

c) The data are described rather than interrogated. What were the authors expecting to find? As the work is not set in the COVID literature, it is presented as though its novelty makes it significant. 

In summary, the literature review needs a substantial overhaul, the method needs explaining in more depth, data re-analysed and discussion will change once stronger arguments have been made to support what was done. 

Author Response

Reviewer 2

Comments and Suggestions for Authors

The research addresses a relevant question and the authors have analysed the data and produced a report on these data. The sample size is sufficiently large to warrant interest. However, the paper could be developed more.

  1. The literature on psychological states and traits experienced during COVID-19 is comprehensive. The authors need to do a review of this literature, and inform the readers on what they did to see what research had been conducted. This is a comprehensive revision.

  • Dear reviewer,

      Thank you for the remark. We tried to comply with your suggestions, which you will see more clearly in the Introduction due to the use of the track changes option.

  1. The authors need to develop a data analysis strategy for what they did. They investigate the validity of measures, which is a good, but do not make a case as to why this should be done and what the impact of this means on the scales themselves and their validity in this context. This could be an important part of the work. 
  • We investigate the validity of the measure to show that it is a relatively stable and acceptable instrument among the university students’ population in the public health context of COVID-19 during the summer of 2022 in the 3.3. “Confirmatory Factor Analysis (CFA)” section. The mean scores are part of the scoring system, which are described through in the 2.3. “Instruments” section in the manuscript. In brief, based on the mean values for each item the overall well-being is categorized from languishing to very high functioning.

  1. The data are described rather than interrogated. What were the authors expecting to find? As the work is not set in the COVID literature, it is presented as though its novelty makes it significant. 
  • Thank you for your remark. Students are exposed to the COVID-19 pandemic in a view of extraordinary social circumstances for a long time, which, based on the literature, in the beginning, resulted in mental health and well-being challenges. As health-related students tend to adapt better to difficult circumstances, we aimed to investigate how the students adapted to these circumstances, more precisely to assess the well-being of Slovenian and Serbian sports science students during the summer of 2022. Our assumption was, that they will have a normal functioning.
  • This research contributes to the previously published empirical research (Pišot et al., 2022), emphasizing the positive responses and successful coping of sports science students in times of difficult and challenging situations, such as the COVID-19 pandemic.

Literature mentioned in the answer:

Pišot, S.; Milovanović, I.M.; Katović, D.; Bartoluci, S.; Radenović, S.S. Benefits of active life in student experiences during COVID-19 pandemic time. Frontiers in Public Health 2022, 10, doi:10.3389/fpubh.2022.971268)

In summary, the literature review needs a substantial overhaul, the method needs explaining in more depth, data re-analysed and discussion will change once stronger arguments have been made to support what was done. 

  • Thank you for your constructive comments. We tried to incorporate all of your suggestions, which you will you hopefully detect while reading the resubmitted manuscript.

Reviewer 3 Report

The study investigated the effect of COVID-19 pandemic on sport sciences students’ well-being. During and following the COVID-19 pandemic, there are numerous studies about effect of it on well-being among numerous populations, therefore, this study does not promote originality. Below you can find my point by point comments.

Abstract: The findings must be written more explicitly to be understood by the reader. The highest factors for what? Factors leading to normal functioning? What did you conclude by implementing this study? What was your suggestions? I hope to find these questions’ answers after reading the manuscript.

Keywords: The title includes all the keywords here, please change them.

Introduction

L54: There are no 2 things requiring you to use “neither”; there are more, so please use “none”.

The introduction was not presented properly (in a harmonious way); I suggest the authors to rewrite it to justify the need of this study with supporting with the literature why this study is important to be published and what message it will convey at the end.

Materials and Methods

L102-107: I did not understand why you applied the questionnaire to 1060 and then put forward a criterion, i.e. sport sciences students, and cut down the number 70%. 

 L126-127: Was this questionnaire adopted into Slovenian and Serbian?  

Results

You indicated on line 96 that 48% of the participants are male but on line 161 52% of the participants are stated to be male. Please check it out.

L233: probably “form” not “from”

The discussion was well-written and handled the findings.  

Author Response

Reviewer 3

Comments and Suggestions for Authors

The study investigated the effect of COVID-19 pandemic on sport sciences students’ well-being. During and following the COVID-19 pandemic, there are numerous studies about effect of it on well-being among numerous populations, therefore, this study does not promote originality. Below you can find my point by point comments.

Abstract: The findings must be written more explicitly to be understood by the reader. The highest factors for what? Factors leading to normal functioning? What did you conclude by implementing this study? What was your suggestions? I hope to find these questions’ answers after reading the manuscript.

  • Dear reviewer,

      Thank you for your comments. We rewrite the “Abstract” for better clarity and understanding, which you will notice since the track changes option was used. With this research, we attempted to investigate the sports sciences students’ well-being in the Summer of 2022. It revealed normal functioning. Furthermore, this research contributes to the previously published research (Pišot et al., 2022), emphasizing the positive responses and successful coping of sports science students in times of difficult and challenging situations, such as the COVID-19 pandemic.

Keywords: The title includes all the keywords here, please change them.

  • We modified the key words as you suggested. These are new keywords: sport; health science; physical education; university students; COVID-19 pandemic; mental health; well-being; PERMA.“

Introduction

L54: There are no 2 things requiring you to use “neither”; there are more, so please use “none”.

  • Thank you for your comment. We changed the mentioned words.

The introduction was not presented properly (in a harmonious way); I suggest the authors to rewrite it to justify the need of this study with supporting with the literature why this study is important to be published and what message it will convey at the end.

  • Thank you for the suggestion. As you will notice while reading The introduction, we have rewrite it, in line with your remarks.

Materials and Methods

L102-107: I did not understand why you applied the questionnaire to 1060 and then put forward a criterion, i.e. sport sciences students, and cut down the number 70%. 

  • Thank you for the remark. This manuscript is part of a more comprehensive research, which included 1060 students of different educational profiles.Given that the "Sports" journal is by definition more focused on sports activities, we decided to single out students of sports sciences for the needs of the manuscript.Furthermore, we explained the research project within the whole section 2. “Materials and Methods”, which will be obvious while reading the mentioned part of the manuscript.

 L126-127: Was this questionnaire adopted into Slovenian and Serbian?  

  • The PERMA-Profiler instrument was used across multipe nations (Butler and Kern, 2016, Avsec et al, 2022; Umucu et al., 2020; Wammerl et al., 2019; Ryan et al., 2019). Specifically, in Slovenia as we can see at Avsec et al. (2022) and in Hungary (Varga et al., 2022) and Croatia Žibrat et al. (2021) wich are teritorically and culturally similar contexts, but no relevant published information found for Serbia. Probably, due to the fact that the questionnaire was published, and the Covid-19 pandemic appeared in recent years, there was a short period of time for collecting research with the sorten verison of the PERMA-Profiler questionnaire in these circumstances and territories.

Literature mentioned in the answer:

  • Butler, J.; Kern, M.L. The PERMA-Profiler: A brief multidimensional measure of flourishing. International Journal of Wellbeing 2016, 6, 1-48, doi:10.5502/ijw.v6i3.526
  • Avsec, A.; Eisenbeck, N.; Carreno, D.F.; Kocjan, G.Z.; Kavčič, T. Coping styles mediate the association between psychological inflexibility and psychological functioning during the COVID-19 pandemic: A crucial role of meaning-centered coping. Journal of contextual behavioral science 2022, 26, 201-209
  • Wammerl, M.; Jaunig, J.; Mairunteregger, T.; Streit, P. The German Version of the PERMA-Profiler: Evidence for Construct and Convergent Validity of the PERMA Theory of Well-Being in German Speaking Countries. Journal of Well-Being Assessment 2019, 3, 75-96, doi:10.1007/s41543-019-00021-0
  • Ryan, J.; Curtis, R.; Olds, T.; Edney, S.; Vandelanotte, C.; Plotnikoff, R.; Maher, C. Psychometric properties of the PERMA Profiler for measuring wellbeing in Australian adults. PLOS ONE 2019, 14, e0225932, doi:10.1371/journal.pone.0225932.
  • Avsec, A.; Kavčič, T.; Kocjan, G.Z. Psychology of Subjective Well-Being; Ljubljana University Press, Faculty of Arts: Slovenia, 2022.
  • Varga, B.A.; Oláh, A.; Vargha, A. A magyar nyelvű PERMA Jóllét Profil kérdőív megbízhatóságának és érvényességének vizsgálata. Mentálhigiéné és Pszichoszomatika 2022, 23, 33-64, doi:10.1556/0406.23.2022.001
  • Žibrat, K. Validation of the PERMA questionnaire. In Proceedings of the 22nd Psychology Days in Zadar, Zadar, 2021.
  • Umucu, E.; Wu, J.-R.; Sanchez, J.; Brooks, J.M.; Chiu, C.-Y.; Tu, W.-M.; Chan, F. Psychometric validation of the PERMA-profiler as a well-being measure for student veterans. Journal of American College Health 2020, 68, 271-277,doi:10.1080/07448481.2018.1546182.

Results

You indicated on line 96 that 48% of the participants are male but on line 161 52% of the participants are stated to be male. Please check it out.

  • Thank you for your observation. We corrected the numbers accordingly to the study sample in the 2.2 “Participants” section. The total sample or this study considered 350 students, of which 182 (52%) were male and 168 (48%) were female.

L233: probably “form” not “from”

  • That is right, thank you. We corrected the word form” not “from”.

The discussion was well-written and handled the findings.  

  • Thank you for your constructive comments.

Reviewer 4 Report

General comments:

Title

The title ends up not providing information that allows identifying what was carried out in the study.

Abstract

The abstract was not written according to the authors' instructions, that is, in a structured way.

The first part of the abstract uses non-colloquial language in scientific studies, such as “etc”, in addition, the Background must summarize the introduction of the manuscript, which does not occur. The Background must be overwritten.

The methodology should be better explained and the results should be presented including absolute and statistical values.

The conclusion should bring the practical applications of the findings.

Please confirm that the keywords are found as descriptors in health sciences.

Introduction

This is very extensive, and on the other hand methodologically explains some points that should this in methodology and not in the introduction.

It is suggested that the introduction have between three to five paragraphs, that they can present the epidemiological context at the beginning and link it with the question under study, which does not occur.

The introduction is not starting from general to specific. It should initially present a more general approach and gradually address the problem (gap) and then present the objective.

The introduction ends up raising other demands that are not the objective of the study. The objective of the study is to evaluate the repercussion of the pandemic and the well-being of sports science students. If a sports science problem with the pandemic is not found, the manuscript is outside the scope of the journal. It is noteworthy that the journal aims to investigate “important topics which are relevant to sport sciences and public health”. That is, the problem must be related to the aforementioned.

From the moment the problem is identified, studies that are controversial in relation to the subject are presented, at this moment the objectives would be justified and after the objectives there should be hypotheses to be answered by the study.

Methods

It should present more clearly the design of the study. A CONSORT or time line, should be presented in order to get a better view of the study design.

The methodology should be divided into five points, design, participants, instruments, procedures and statistics.

The sample should be better explained with the number of subjects presented initially and then present the inclusion and exclusion criteria. How was the number of participants presented arrived at, since the manuscript mentions that this number was much higher. Was any statistical test performed, or was any statistical program used for this? And the ethical procedures used? The Declaration of Helsinki mentioned was in relation to the addendum of which year?

Regarding the instrument used, it should be better described, with a summary and even the cutoff points used. It must be presented who conceived the instrument and who validated it for the countries where the research was carried out, with the due references. If not validated, was there at least validation by judges regarding possible suitability for the local language? This instrument is aimed at this audience, what is the reference for this?

In the procedures, it should be mentioned how the questionnaire was applied, who participated and how, if any training was used and even if any correlation was made between the applicators, if any. If it was self-administered, does the instrument validation allow for this? Please clarify.

Statistical treatment should be better detailed in order to better follow what has been done. Measures of central tendency and all data referring to the aforementioned calculation should be presented. In the correlation there should be “p” values.

Bearing in mind that the study intends to carry out a confirmatory factor analysis, this should be carried out by comparing the results with an already validated instrument. Not having this instrument and still dealing with a translation, wouldn't it be feasible to do an exploratory factor analysis initially and then carry out the confirmatory one? Please comment, preferably with some reference to the proposal.

Please consult Cohen (1988).

Results

Are presented satisfactorily. However, after adequacy as mentioned in the methodology, some data must be adequate.

Discussion

It should reaffirm the objectives and start discussing the results in the chronological order that appear in the item results.

The discussion is very short, smaller than the introduction, even in the face of so much data presented. It would be important to discuss the data more exhaustively in light of the literature.

Furthermore, the limitations are also superficial, given what is mentioned in the review. It is important that the discussion presents studies for and against the findings and that it seeks a possible explanation for the results found, thus explaining the data.

Conclusion

The conclusion is in line with the guidelines of the journal, which should be focused on the findings and even with practical applications for the results found, responding to the objectives proposed by the study.

References

The formatting of the references must be revised according to the instructions to the authors, and even of the 40 references, 32 are current and the rest have more than five years of publication, which demonstrates a current and satisfactory theoretical framework.

Overview

The manuscript presented addresses a relevant research topic.

It would be advisable to do a general review.

Author Response

Reviewer 4

Comments and Suggestions for Authors

General comments:

Title

The title ends up not providing information that allows identifying what was carried out in the study.

  • Dear reviewer,

      Thank you for your comments. In accordance, we made changes in the manuscript title into: “Exploring the Well-being among Slovenian and Serbian Sport Science Students During the COVID-19 Pandemic of Summer 2022”.

Abstract

The abstract was not written according to the authors' instructions, that is, in a structured way.

  • Thank you for this In line with the journal instructions we did not make seperate headings, but we rewrite the abstract for improving and better understanding of the current study.

Part of the authors’ instruction: "Abstract: The abstract should be a total of about 200 words maximum. The abstract should be a single paragraph and should follow the style of structured abstracts, but without headings: 1) Background: Place the question addressed in a broad context and highlight the purpose of the study; 2) Methods: Describe briefly the main methods or treatments applied. Include any relevant preregistration numbers, and species and strains of any animals used. 3) Results: Summarize the article's main findings; and 4) Conclusion: Indicate the main conclusions or interpretations. The abstract should be an objective representation of the article: it must not contain results which are not presented and substantiated in the main text and should not exaggerate the main conclusions..."

The first part of the abstract uses non-colloquial language in scientific studies, such as “etc”, in addition, the Background must summarize the introduction of the manuscript, which does not occur. The Background must be overwritten. The methodology should be better explained and the results should be presented including absolute and statistical values. The conclusion should bring the practical applications of the findings.

  • Thank you for these remarks. We rewrite the whole “Abstract” section of the manuscript, with the implementation of your suggestions for better clarity and understanding.

Please confirm that the keywords are found as descriptors in health sciences.

  • We modified the key words in the present study you suggested: “sport; health science; physical education; university students; COVID-19 pandemic; mental health; well-being; PERMA”.

Introduction

This is very extensive, and on the other hand methodologically explains some points that should this in methodology and not in the introduction.

  • Thank you for your We dislocated the PERMA-Profiler questionnaire explanation to the 2. “Material and Methods” section.

It is suggested that the introduction have between three to five paragraphs, that they can present the epidemiological context at the beginning and link it with the question under study, which does not occur.

  • Thank you for the constructive suggestion. You will notice that we have implemented it.

The introduction is not starting from general to specific. It should initially present a more general approach and gradually address the problem (gap) and then present the objective.

  • Thank you for the constructive suggestion. We tried to rewrite The introduction in line with this suggestion of yours.

The introduction ends up raising other demands that are not the objective of the study. The objective of the study is to evaluate the repercussion of the pandemic and the well-being of sports science students. If a sports science problem with the pandemic is not found, the manuscript is outside the scope of the journal. It is noteworthy that the journal aims to investigate “important topics which are relevant to sport sciences and public health”. That is, the problem must be related to the aforementioned.

  • Thank you for the remark. After reading the Aim and scope section of the “Sports” journal, we believe that the manuscript is inside of the scope of the journal.

From the moment the problem is identified, studies that are controversial in relation to the subject are presented, at this moment the objectives would be justified and after the objectives there should be hypotheses to be answered by the study.

  • Thank you for your comment. As you suggested, we made corrections to the structure of the “Introduction” section. Our hypothesis was presented “Therefore, we assume that sport science students adapted well and have normal functioning in their everyday lives during the first waves of COVID-19 pandemic.”. Following the research, the results are aligned with our assumption, and we can accept our hypothesis for this study.

Methods

It should present more clearly the design of the study. A CONSORT or time line, should be presented in order to get a better view of the study design.

  • Thank you for the valuable suggestion. We added a time line as a Figure 1. in the manuscript, within the section 2.2. “Participants”.

The methodology should be divided into five points, design, participants, instruments, procedures, and statistics.

  • As you suggested, the methodology section is divided into five points: 2.1. Design, 2.2. Participants, 2.3. Instruments, 2.4. Procedures and 2.5. Statistics.

The sample should be better explained with the number of subjects presented initially and then present the inclusion and exclusion criteria. How was the number of participants presented arrived at, since the manuscript mentions that this number was much higher. Was any statistical test performed, or was any statistical program used for this? And the ethical procedures used? The Declaration of Helsinki mentioned was in relation to the addendum of which year?

  • Thank you for these comments. We described in more detail the bigger research project in the 2.1. “Design” section, and the sampling process in the 2.2 “Participants” section. We did not used any statistical analysis for the sampling process. The data collecting and ethical procedures are described through the 2.4 “Procedures” section. The students before participating in this study had to agree in a form of consent for participating in this study voluntary and anonymously at the beginning of the questionnaire. They could withdraw at any time. Their data are protected and only used for research purposes. Ethical approval prior to data collection last year was obtained at the Faculty of Sport and Physical Education, University of Novi Sad (No. 47-12-12/2021-1).

Regarding the instrument used, it should be better described, with a summary and even the cutoff points used. It must be presented who conceived the instrument and who validated it for the countries where the research was carried out, with the due references. If not validated, was there at least validation by judges regarding possible suitability for the local language? This instrument is aimed at this audience, what is the reference for this?

  • Regarding to the instrument, we described in more depth in the “Materials and Methods” section under the 2.3 “Instruments” subheading. The scoring process, the cutoff points are described at the methods section in the 2.3 “Instruments” subheading.
  • Furthermore, we explained the background theory, looking back for Martin Seligman’s Well-being theory from positive psychology, which includes five dimensions. Those dimensions (positive emotion, engagement, relationship, meaning and accomplishment) together make the PERMA acronym. Butler and Keln made a tool for measuring well-being using the PERMA constructs. The instrument validation process can be found in the following literature: Butler and Kern (2016). In brief, the English version was tested for its psychometric measures across a large international sample (N=31.966) and resulted in a 15 items questionnaire. In addition, 8 items for assessing negative emotions were added. The validation study of this measurement tool indicated adequate reliability and validity of the scale. (Butler and Kern, 2016). Therefore, based on the previously published research across many countries and sample using both the 23 item instrument and the shortened version of 15 items, we decided to implicate the sorter version of the questionnaire to our study (Avsec et al, 2022; Žibrat et al. 2021, Avsec et al., 2022; Umucu et al., 2020; Wammerl et al., 2019; Ryan et al., 2019; Varga et al., 2022). It was more convenient to add for the shorten version of survey for the comprehensive research project. Furthermore, we investigate the validity of the measure to show that it is a relatively stable and acceptable instrument among the university students’ population in the public health context of COVID-19 during the summer of 2022 in the 3.3. “Confirmatory Factor Analysis (CFA)” section.
  • In addition, the PERMA-Profiler instrument was used in Slovenia as we can see at Avsec et al. (2022) and in Hungary (Varga et al., 2022) and Croatia (Žibrat et al., 2021) wich are teritorically and culturally similar contexts, but no relevant published information found for Serbia. Probably, due to the fact that the questionnaire was published, and the Covid-19 pandemic appeared in recent years, there was a short period of time for collecting research with the sorten verison of the PERMA-Profiler questionnaire in these circumstances and territories.

Literature mentioned in the answer:

  • Butler, J.; Kern, M.L. The PERMA-Profiler: A brief multidimensional measure of flourishing. International Journal of Wellbeing 2016, 6, 1-48, doi:10.5502/ijw.v6i3.526
  • Avsec, A.; Eisenbeck, N.; Carreno, D.F.; Kocjan, G.Z.; Kavčič, T. Coping styles mediate the association between psychological inflexibility and psychological functioning during the COVID-19 pandemic: A crucial role of meaning-centered coping. Journal of contextual behavioral science 2022, 26, 201-209
  • Umucu, E.; Wu, J.-R.; Sanchez, J.; Brooks, J.M.; Chiu, C.-Y.; Tu, W.-M.; Chan, F. Psychometric validation of the PERMA-profiler as a well-being measure for student veterans. Journal of American College Health 2020, 68, 271-277, doi:10.1080/07448481.2018.1546182.
  • Wammerl, M.; Jaunig, J.; Mairunteregger, T.; Streit, P. The German Version of the PERMA-Profiler: Evidence for Construct and Convergent Validity of the PERMA Theory of Well-Being in German Speaking Countries. Journal of Well-Being Assessment 2019, 3, 75-96, doi:10.1007/s41543-019-00021-0
  • Ryan, J.; Curtis, R.; Olds, T.; Edney, S.; Vandelanotte, C.; Plotnikoff, R.; Maher, C. Psychometric properties of the PERMA Profiler for measuring wellbeing in Australian adults. PLOS ONE 2019, 14, e0225932, doi:10.1371/journal.pone.0225932.
  • Avsec, A.; Kavčič, T.; Kocjan, G.Z. Psychology of Subjective Well-Being; Ljubljana University Press, Faculty of Arts: Slovenia, 2022.
  • Varga, B.A.; Oláh, A.; Vargha, A. A magyar nyelvű PERMA Jóllét Profil kérdőív megbízhatóságának és érvényességének vizsgálata. Mentálhigiéné és Pszichoszomatika 2022, 23, 33-64, doi:10.1556/0406.23.2022.001
  • Žibrat, K. Validation of the PERMA questionnaire. In Proceedings of the 22nd Psychology Days in Zadar, Zadar, 2021.

In the procedures, it should be mentioned how the questionnaire was applied, who participated and how, if any training was used and even if any correlation was made between the applicators, if any. If it was self-administered, does the instrument validation allow for this? Please clarify.

  • Thank you for your comment. As we mentioned in the 2.1. “Design” section, this study is a part of a more comprehensive research project. As you proposed, we described in more detail the sampling process in the 2.2. “Participants” section, the instrument description in the 2.3. “Instruments” section, and the data collection process in the 2.4. “Procedures” section. We did not have any training included.

Statistical treatment should be better detailed in order to better follow what has been done. Measures of central tendency and all data referring to the aforementioned calculation should be presented. In the correlation there should be “p” values.

  • Thank you for your comment. We described in more detail the statistical treatment in the 2. “Materials and Methods” section above the 2.5 “Statistics subsection” and the 3. “Results” section of the manuscript. „p” values have been stated in the correlation (3. Results).

Bearing in mind that the study intends to carry out a confirmatory factor analysis, this should be carried out by comparing the results with an already validated instrument. Not having this instrument and still dealing with a translation, wouldn't it be feasible to do an exploratory factor analysis initially and then carry out the confirmatory one? Please comment, preferably with some reference to the proposal.

  • Comparison obtained results with already validated instrument showed that the biggest difference is in Engagement, where the authors who validated the instrument calculated this scale with α =0.72 (Butler, & Kern, 2016), while confirmatory factor analysis in this paper revealed α =0.54. In all other scales, we didn’t determine important differences in validity (only from 0.01 to 0.05).

Results

Are presented satisfactorily. However, after adequacy as mentioned in the methodology, some data must be adequate.

  • Dear reviewer,

since we have made changes to the "Method" section, we believe that the results have been amended according to said changes.

Discussion

It should reaffirm the objectives and start discussing the results in the chronological order that appear in the item results.

  • Thank you. We have tried to discuss the results in the chronological order that appear in the item results.

The discussion is very short, smaller than the introduction, even in the face of so much data presented. It would be important to discuss the data more exhaustively in light of the literature.

  • We have expanded the discussion in accordance with your recommendations.

Furthermore, the limitations are also superficial, given what is mentioned in the review.

  • Thank you for your remark. We described the strength and the limitations of the present study more deeply within the second part of the section 4. “Discussion” section.

It is important that the discussion presents studies for and against the findings and that it seeks a possible explanation for the results found, thus explaining the data.

  • Dear reviewer,

we tried to harmonize the discussion with your constructive remark, which you will see by reading the "Discussion" section.

Conclusion

The conclusion is in line with the guidelines of the journal, which should be focused on the findings and even with practical applications for the results found, responding to the objectives proposed by the study.

  • Thank you for your affirmative comment.

References

The formatting of the references must be revised according to the instructions to the authors, and even of the 40 references, 32 are current and the rest have more than five years of publication, which demonstrates a current and satisfactory theoretical framework.

  • The formatting references were made by the available style file for Endnote with the ACS style guide from the authors’ guideline page. While writing the manuscript, we followed the following links: https://www.mdpi.com/journal/sports/instructions and https://www.mdpi.com/authors/references .

Overview

The manuscript presented addresses a relevant research topic.

It would be advisable to do a general review.

  • Thank you for your constructive comments. We tried to fulfill all of them.

Round 2

Reviewer 2 Report

The article has been addressed and now ready more clearly. One issue the authors need to address is the value of establishing validity of the scale in a different language. This is an important aspect of the work and the authors did a comprehensive job examining the validity. There should a section in the literature review on the issue. There have been entire articles dedicated to testing the validity of a scale in a new population and so some discussion is needed. 

Author Response

Dear Reviewer,

Thank you for the time and effort you put into reviewing our manuscript.

Changes and additions, in accordance with your suggestions, can be found in section 2.3. Instruments

Sincerely yours,

Authors

Reviewer 4 Report

In view of the adaptations presented, I consider the manuscript in conditions to be published

Author Response

Dear Reviewer,

Thank you for your comments.